# Adaptive modeling and inference of higher-order coordination in neuronal assemblies: A dynamic greedy estimation approach

**Shoutik Mukherjee**[1,2], **Behtash Babadi**[1,2]*

**1** Department of Electrical and Computer Engineering, University of Maryland, College Park, Maryland, United States of America, **2** Institute for Systems Research, University of Maryland, College Park, Maryland, United States of America

* behtash@umd.edu

**Data Availability Statement:** There are no primary data in the paper. Software implementations in MATLAB v2017b of the algorithms discussed are available at https://github.com/ShoutikM/

## Abstract

Central in the study of population codes, coordinated ensemble spiking activity is widely observable in neural recordings with hypothesized roles in robust stimulus representation, interareal communication, and learning and memory formation. Model-free measures of synchrony characterize coherent pairwise activity but not higher-order interactions, a limitation transcended by statistical models of ensemble spiking activity. However, existing model-based analyses often impose assumptions about the relevance of higher-order interactions and require repeated trials to characterize dynamics in the correlational structure of ensemble activity. To address these shortcomings, we propose an adaptive greedy filtering algorithm based on a discretized mark point-process model of ensemble spiking and a corresponding statistical inference framework to identify significant higher-order coordination. In the course of developing a precise statistical test, we show that confidence intervals can be constructed for greedily estimated parameters. We demonstrate the utility of our proposed methods on simulated neuronal assemblies. Applied to multi-electrode recordings from human and rat cortical assemblies, our proposed methods provide new insights into the dynamics underlying localized population activity during transitions between brain states.

## Author summary

Simultaneous ensemble spiking is hypothesized to have important roles in neural encoding; however, neurons can also spike simultaneously by chance. In order to characterize the potentially time-varying higher-order correlational structure of ensemble spiking, we propose an adaptive greedy filtering algorithm that estimates the rate of all reliably-occurring simultaneous ensemble spiking events. Moreover, we propose an accompanying statistical inference framework to distinguish the chance occurrence of simultaneous spiking events from coordinated higher-order spiking. We demonstrate the proposed methods accurately differentiate coordinated simultaneous spiking from chance occurrences in simulated data. In application to human and rat cortical data, the proposed methods

AdaptiveHigherOrderCoordination and https://www.doi.org/10.5281/zenodo.10009981.

**Funding:** This work is supported in part by the National Science Foundation (nsf.gov) Awards No. CCF1552946 and ECCS2032649 (to BB) and the National Institutes of Health (nih.gov) Award No. 1U19NS107464 (to BB). The funders did not play any role in the study design, data collection and analysis, decision to publish, or preparation of the manuscript.

**Competing interests:** The authors have declared that no competing interests exist.

reveal time-varying dynamics in higher-order coordination that coincide with changing brain states.

## Introduction

Synchronous neuronal ensemble activity is central in the study of neural population codes. Coordinated ensemble spiking has been observed in a variety of brain areas, prompting a variety of hypotheses about its role in cognitive function. For instance, studies have documented synchronous spiking at all levels of the mammalian visual pathway [1–3]. Synchronized thalamic population activity has also been widely observed, a phenomenon to which visual cortical neurons have been found sensitive, suggesting the importance of synchronized neuronal activity in thalamocortical communication [4, 5]. Synchronized spiking has, more broadly, been hypothesized to influence inter-areal communication and the flow of neural information [6–10]. The study of coordinated neural activity is also closely tied to oscillatory activity and memory. Synchronized hippocampal and hippocampal-cortical activity are thought to have significant roles in memory formation, working memory tasks, and encoding information for spatial navigation [11–13]. Coordinated ensemble spiking has additionally been postulated to be mediated by oscillations in local field potentials [14–16].

The prevalence of coordinated spiking and its functional implications for a range of neural processes have motivated both model-free and model-based approaches to quantifying spiking synchrony. Perhaps the most intuitive model-free metric is the pairwise correlations of spike trains smoothed by a Gaussian (or exponential) kernel [17, 18]. Other model-free measures include a range of spike train distance metrics that also perform pairwise comparisons [19]. Though the coherence of pairwise activity can be described, such measures do not capture higher-order coordination, and are limited in the ability to model dynamics in or determine the significance of pairwise coherence without repeated trials. More recently, model-free approaches based on continuous-time transfer entropy formulations have been introduced that avoid smoothing or binning spike trains [20–23]; however these are still limited to pairwise measures of synchrony.

Statistical models of neuronal ensemble activity transcend the limitation of model-free metrics to pairwise comparisons. Two widely used approaches are the maximum entropy models and point process generalized linear models (GLM) [24, 25]. Maximum entropy models describe the state of the neural population only in terms of its instantaneous correlational structure [26, 27]. Models are estimated to match observed firing rates and all pairwise (and potentially higher order) correlations simultaneously. The suitability of the maximum entropy model formulation for analyzing coordinated spiking has motivated several extensions. For instance, Bayesian state-space filtering algorithms have been developed to capture dynamics in the strength of higher-order spiking interactions [28, 29]. A stimulus-dependent maximum entropy model has also been proposed to address potential synchrony-modulating factors [30]. Extensions also include efforts to address the computational complexity associated with analyzing higher-order interactions amongst large neuronal assemblies [31, 32].

Point process GLMs are a common alternative to maximum entropy models for ensemble spiking [33, 34] that can characterize the influence of past population activity and other relevant covariates. Though useful in estimating functional connectivity [35, 36], each neuron must be assumed conditional independent due to regularity conditions that prohibit simultaneous spiking events [37–39]. This can be circumvented by using an equivalent marked point processes (MkPP) representation that explicitly models each disjoint simultaneous spiking

event [38]. MkPP representations of ensemble activity have also been utilized to analyze neuronal population coding in unsorted spiking data [40, 41]. A related approach models disjoint simultaneous spiking events as log-linear combinations of point process models that permits an intuitive representation of excess or suppressed synchrony [15, 39].

The aforementioned statistical models enable the analysis of higher-order coordination in ensemble spiking, though each with their respective limitations. Dynamics in the correlational structure of maximum entropy models may be tracked with state-space filtering algorithms and credible intervals can be constructed to assess the statistical significance of correlations; however, the influence of past population activity on the ensemble state is neglected, and assumptions on the relevance of higher-order interactions are typically imposed for tractability of model estimation. Log-linear point process models can track dynamics in coherent spiking while incorporating the effects of temporal dynamics of population activity, and confidence intervals may be approximated for statistical inference; the necessity of *a priori* assumptions on the relevance of higher-order interactions still, however, remains. Additionally, both approaches require multiple repeated trials to capture dynamics in correlational structure with statistical confidence, thus limiting their applicability to spiking data without trial structure. A discretized MkPP model is capable of capturing greater detail in the effects of past population activity on coordinated spiking, though these effects are assumed to be static. To our knowledge, the tractability of the MkPP model for ensemble spiking has not been addressed, and a corresponding statistical inference framework is lacking.

We address these gaps by proposing an adaptive greedy filtering algorithm based on the discretized MkPP formulation in [38] to model dynamics in higher-order spiking coordination in single-trial recordings while capturing the influence of past ensemble activity. Incorporating similar data-driven restrictions on modeled interactions as in [31], we also address the question of tractability of the discretized MkPP formulation. Furthermore, we build on recent theoretical results related to Adaptive Granger Causality (AGC) analysis [35] to provide a precise statistical framework to detect significant coordinated spiking activity of arbitrary order. We demonstrate our proposed method's utility in tracking dynamics in synchronous activity with statistical confidence on simulated ensemble spiking. Applying our method to continuous multi-electrode recordings of human cortical assemblies during anesthesia and to rat cortical assemblies during sleep provides novel insights into coordinated spiking dynamics that underlie transitions between brain states.

## Materials and methods

In the following, we first highlight the limitations of existing approaches in application to neuronal spiking data without trial structure in order to motivate and highlight the contributions of this work, namely a framework for the dynamic and statistically precise inference of latent coordinated spiking in neuronal assemblies using their simultaneous spiking representation (Fig 1, bottom panel). We summarize essential components of the proposed methods subsequently. Key notation used throughout the remainder of the paper is summarized in Table 1. Problem formulation, algorithm development, and theoretical results are comprehensively addressed in supporting information (S1 Appendix). Software implementations in MATLAB v2017b of the algorithms discussed here are available at https://github.com/ShoutikM/AdaptiveHigherOrderCoordination and https://www.doi.org/10.5281/zenodo.10009981.

### Related work

In the context of existing approaches to the analysis of correlational structure underlying ensemble spiking activity, the methods proposed in this work address two key limitations. The

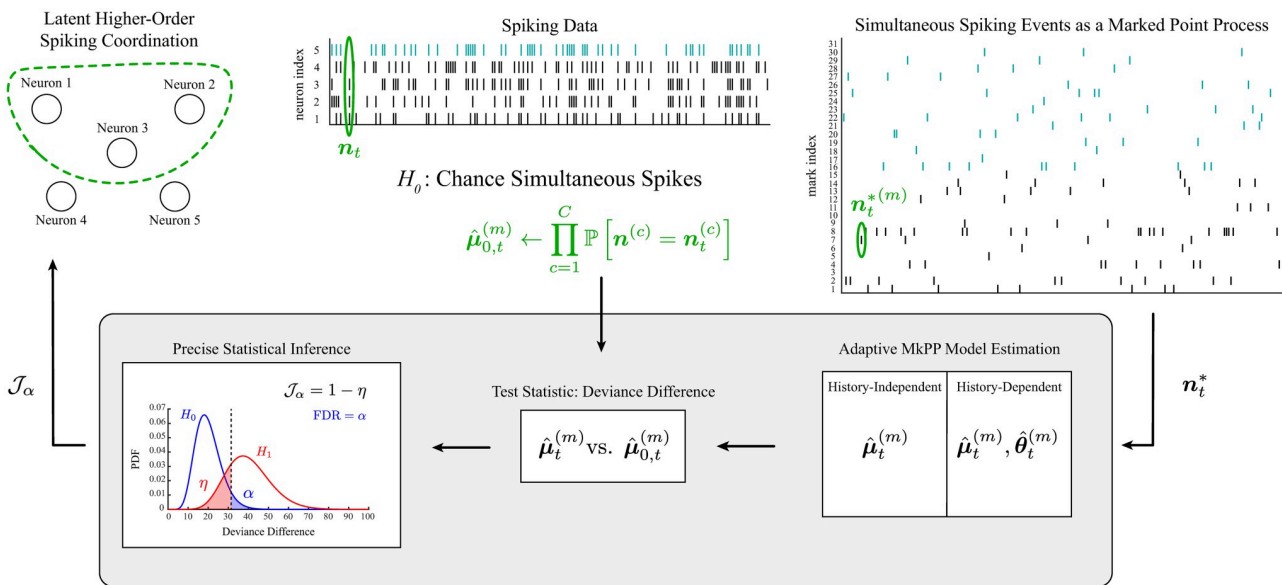

**Fig 1. Summary of methods.** In order to characterize latent higher-order spiking coordination in a group of $C$ neurons, the observed spiking event at every time frame $t$, denoted $n_t$, is mapped to a categorical vector $n_t^*$ that indicates which of $2^C - 1$ possible simultaneous spiking events occurred. The mapping of one spiking outcome to its categorical representation is circled in green. The disjointness of the categorical representation means that the spiking activity of a neuron can be recovered as the sum of outcomes in which that neuron spiked (as indicated in teal for neuron 5). In the proposed methods (grey box), this disjoint representation of spiking outcomes is modeled as a marked point process (MkPP). A time-varying categorical distribution describing the probability of each category (simultaneous spiking event) is fit by adaptively estimating each's base-rate parameter ($\mu_t^{(m)}$), and possibly the influence of the recent ensemble spiking history ($\theta_t^{(m)}$). To determine if $r^{th}$-order events (simultaneous spiking of exactly $r$ neurons) occur at rates differing significantly from the rate of chance $r^{th}$-order events, the estimated base-rate parameters of all $r^{th}$-order events are compared to values of base-rate parameters expected if the neurons were independent. Significant $r^{th}$-order coordination is indicated by rejecting a null hypothesis that such events occur by chance in favor of the alternative that they occur at a significantly different rate using a statistical test with precisely characterized distributions for the test statistic under both hypotheses. This framework enables the statistical confidence of $r^{th}$-order coordination detected at a false discovery rate, $\alpha$, to be summarized in terms of type I and type II errors using Youden's $J$-statistic, $\mathcal{J}_\alpha$.

**Table 1. Summary of key notation.**

| Notation | Definition |
|---|---|
| $\boldsymbol{n}_t = [n_t^{(1)}, \ldots, n_t^{(C)}]^\top$ | Ensemble spiking observation at time bin $t$ of $C$ neurons |
| $\lambda_t^{(c)}\Delta = \mathbb{P}[n_t^{(c)} = 1\|\mathcal{H}_t]$ | Conditional Intensity Function (CIF) of $c^{th}$ neuron |
| $\boldsymbol{n}_t^* = [n_t^{*(1)}, \ldots, n_t^{*(C^*)}]^\top$ | Marked observations at time bin $t$ of $C^* = 2^C - 1$ marks |
| $\lambda_t^*(m)\Delta = \mathbb{P}[n_t^*(m) = 1\|\mathcal{H}_t]$ | CIF of $m^{th}$ mark |
| $n_t^{(g)}$ | Ground process, $\sum_{m=1}^{C^*} n_t^{*(m)}$ |
| $\lambda_t^{*(g)}\Delta$ | CIF of the ground process, $\sum_{m=1}^{C^*} \lambda_t^{*(m)}\Delta$ |
| $\boldsymbol{\omega}_t = [\boldsymbol{\omega}_t^{(1)^\top}, \boldsymbol{\omega}_t^{(2)^\top}, \ldots, \boldsymbol{\omega}_t^{(C^*)^\top}]^\top$ | Model parameters of history-dependent model |
| $\boldsymbol{\mu}_t = [\mu_t^{(1)}, \ldots, \mu_t^{(C^*)}]^\top$ | Base rate parameters of marks |
| $\boldsymbol{\theta}_t^{(m)}$ | History-dependence coefficients of the $m^{th}$ mark |
| $u_t^{(m)}$ | Log-odds of $m^{th}$ mark vs. no spiking |
| $u_{0,t}^{(m)}$ | Log-odds of $m^{th}$ mark vs. no spiking (reduced model) |
| $\gamma_t^{(m)} = u_t^{(m)} - u_{0,t}^{(m)}$ | Exogenous factor for $m^{th}$ mark |
| $\beta$ | Forgetting factor, $0 < \beta < 1$ |
| $W$ | Window length |

first is the consideration of simultaneous spiking between more than 2 neurons. Existing model-free approaches are largely restricted to pairwise evaluations of synchrony. Intuitive metrics like correlations or other similarity measures compare pairs of smoothed spike trains [17–19]. Recently, model-free analyses based on continuous-time formulations of transfer entropy have been proposed to evaluate causal history-dependent interactions and synchronization [20–23]. While not reliant on binning or smoothing spiking data, the tests of conditional independence in [20, 21] examine the relationship between the history of one point process and the updates of another, thus resembling Granger causality analysis [35, 36] more closely than a measure of synchrony. Alternatively, the transfer entropy-based approach in [22, 23] computes the mutual information rate as the sum of transfer entropy rates between two point process in both direction. Though this symmetric notion of connectivity quantifies synchronization, it remains limited to pairwise interactions.

Principal component analysis (PCA) and independent component analysis (ICA) have also been used to identify groups of synchronously spiking neurons [42–45], using multiple trials to form empirical covariance matrices and thereby discover a low-dimensional set of activity sources. Noting that PCA uses the covariance matrix to identify the low-dimensional components, it shares the same limitations with other pairwise measures, when the latent correlational structure is of higher order. While ICA can identify sources that account for higher-order correlations [43, 45], neither of these methods is well-suited to single-trial spike trains or when the underlying data structure is not low-dimensional or stationary.

While model-based approaches can circumvent the limitation to pairwise synchronization, in the same fashion as ICA, they are ill-suited to capturing dynamics in spiking data absent trial structure with statistical confidence. In the cases of two approaches closely related to the proposed methods, we examine this limitation more closely. We first consider the approach in [28], which utilizes a maximum entropy model of ensemble spiking whose dynamics are captured using Bayesian state-space filtering. The method, tailored for multi-trial data, assumes stationarity across trials but non-stationarity within so that the likelihood of the spiking observations $\boldsymbol{n}_t$ is expressed as

$$p(\boldsymbol{n}_{t=1:T,l=1:L}) = \prod_{t,l} \exp\left(\sum_i \theta_t^{(i)} n_{t,l}^{(i)} + \sum_{i<j} \theta_t^{(i,j)} n_{t,l}^{(i)} n_{t,l}^{(j)} + \cdots + \theta_t^{(1,\cdots,C)} n_{t,l}^{(1)} \cdots n_{t,l}^{(C)} - \psi(\boldsymbol{\theta}_t)\right),$$

with $T$ time bins and $L$ trials, and parameters evolving according to the linear-Gaussian state transition $\boldsymbol{\theta}_t = \boldsymbol{F}\boldsymbol{\theta}_{t-1} + \phi_t$. The Bayesian state-space filter utilizes an expectation-maximization algorithm that requires evaluating the posterior density $p(\boldsymbol{\theta}_{t=1:T}|\boldsymbol{n}_{t=1:T,l=1:L})$. Assuming non-stationarity in the trial-free case, the posterior, which is also used to construct credible intervals, is poorly estimated.

Next, we examine the log-linear point process model approach in [39]. Here, the log-probability of a simultaneous spiking event is modeled as the sum of log-probabilities of conditionally independent neurons and an additional term that captures excess synchrony. As an example, the joint probability for a pair of neurons modeled as

$$\log \mathbb{P}[n_t^{(1)}, n_t^{(2)}] = n_t^{(1)} \log \lambda_t^{(1)} + n_t^{(2)} \log \lambda_t^{(2)} + n_t^{(1)} n_t^{(2)} \xi_t^{(1,2)}.$$

Significant synchronization is detected if the null hypothesis that $\xi_t^{(1,2)} = 0$ can be rejected. However, doing so requires a bootstrap distribution for $\xi_t^{(1,2)}$, an ill-posed proposition in the trial-free case if the term is allowed to be time-varying.

The proposed method uses a discretized marked point process model (MkPP) of ensemble spiking activity [38] to capture higher-order interactions. The contributions of this work are the development of a greedy approach to adaptively fit the MkPP model to capture dynamics in spiking data without trial structure and a statistical inference framework with precisely

characterized distributions for the test statistic, thus circumventing the need for bootstrapping or estimating a posterior distribution, both of which are better suited to multi-trial data when time-varying higher-order coordination is considered.

## Discretized marked point process likelihood model

To characterize coordinated spiking, it is necessary to use an appropriate representation of neuronal ensemble spiking. Because multivariate point processes as defined in literature [37] do not permit simultaneous events at arbitrarily small time scales, point process models of ensemble spiking treat neurons as conditionally independent elements of a multivariate process. Instead, we use a discrete-time marked point process (MkPP) model to avoid assuming conditional independence, disjointly representing all simultaneous spiking outcomes [38, 39, 46].

For an assembly of $C$ neurons, the $C$-variate spiking process, binned with small bin size $\Delta$, at time bin index $t$ is denoted by $\boldsymbol{n}_t$. Rather than treating its components as conditionally independent, $\boldsymbol{n}_t$ is viewed as a multivariate observation that we map to a $C^*$-variate process, a MkPP whose marks count exactly on of $C^* := 2^C - 1$ possible non-zero spiking events. That is, at each time $t_j$ such that $\boldsymbol{n}_{t_j} \neq \boldsymbol{0}$, the sole non-zero element of $\boldsymbol{n}_{t_j}^*$ indicates which mark (ensemble spiking outcome) has occurred. We also define the binned ground process $n_t^{(g)} := \sum_{m=1}^{C^*} n_t^{*(m)}$ that indicates the occurrence of any spiking event [37]. The mark space $\mathcal{K} := \{1, \ldots, C^*\}$ indexes the non-zero spiking outcomes [37]. For example, Fig 1 shows the activity of $C = 5$ neurons mapped to a marked process with $C^* = 31$ marks; the event that neurons 1–3 spike together while 4 and 5 do not maps to the mark index 7 (circled in green). Because the marked representation is disjoint, the spiking observations of neuron $c$ can be recovered by adding the observations of all marks corresponding to events including neuron $c$ spiking. For instance, in Fig 1, the spiking activity of neuron 5 is the sum of simultaneous spiking events indexed 16–31 (both indicated in teal).

The conditional intensity functions (CIFs) of $\boldsymbol{n}_t$ and $\boldsymbol{n}_t^*$ are approximated by the probabilities of observing an event at time bin $t$ given ensemble spiking history; they are denoted $\lambda_t^{(c)}\Delta$ and $\lambda_t^{*(m)}\Delta$ for $c = 1, \ldots, C$ and $m = 1, \ldots, C^*$, respectively. We can relate $\lambda_t^{(c)}\Delta$ to $\lambda_t^{*(m)}\Delta$ in the same manner as $n_t^{(c)}$ to $n_t^{*(m)}$, and obtain the CIF of the ground process $\lambda_t^{*(g)}\Delta = \sum_{m=1}^{C^*} \lambda_t^{*(m)}\Delta$.

The joint distribution of the MkPP is expressed as a multinomial generalized linear model (mGLM) with multinomial logistic link function, two versions of which are considered here. The first, more general version utilizes the ensemble history as covariates in the mGLM. Here, the log-odds of the $m^{\text{th}}$ mark occurring, i.e. the probability of observing the $m^{\text{th}}$ simultaneous spiking outcome $\lambda_t^{*(m)}\Delta$, versus no spiking event occurring are:

$$\boldsymbol{x}_t^\top \boldsymbol{\omega}_t^{(m)} = \log\left(\frac{\lambda_t^{*(m)}\Delta}{1 - \lambda_t^{*(g)}\Delta}\right). \tag{1}$$

The log-odds of the $m^{\text{th}}$ mark are parameterized by $\boldsymbol{\omega}_t^{(m)} := [\mu_t^{(m)}, \boldsymbol{\theta}_t^{(m)\top}]^\top$, consisting of an ensemble history-modulation vector $\boldsymbol{\theta}_t^{(m)}$ and the baseline firing parameter, $\mu_t^{(m)}$; $\boldsymbol{x}_t$ denotes the recent ensemble spiking history.

The second version of the mGLM makes the simplifying assumption that there is no history dependence. Consequently, the history-independent model is parameterized only by the baseline firing parameters $\boldsymbol{\mu}_t = [\mu_t^{(1)}, \mu_t^{(2)}, \ldots, \mu_t^{(C^*)}]^\top$. Here, the log-odds of the $m^{\text{th}}$ mark occurring

versus no spiking event occurring are:

$$\mu_t^{(m)} = \log\left(\frac{\lambda_t^{*(m)}\Delta}{1 - \lambda_t^{*(g)}\Delta}\right). \tag{2}$$

Summarily, the discretized MkPP model represents ensemble spiking activity explicitly in terms of all possible outcomes to model their likelihoods jointly. Hence, the MkPP model is essentially a time-varying categorical distribution where the categories are the simultaneous spiking outcomes; the instantaneous odds of these outcomes are non-stationary and possibly dependent on recent ensemble activity history.

## Adaptive estimation of marked point process models

Since the parameters of the MkPP mGLM are assumed to be time-varying, we use adaptive algorithms to capture the dynamics of the history-dependent and history-independent models. However, analyzing large neuronal assemblies raises the issue of tractability since the number of parameters to be estimated scales exponentially with $C$. Since it is likely that some marks will not contain any events, we employ a thresholding rule similar to [31], considering only "reliable interactions", i.e. the subset of the mark space $\bar{\mathcal{K}} = \{m \in \mathcal{K} : \sum_t n_t^{*(m)} > N_{thr}\}$ for some pre-defined constant $N_{thr} > 0$, and treating the rates of the remaining marked processes as negligible due to their infrequency. That is, the time-varying categorical distribution assumes the probabilities of categories that occur with negligible frequency are zero so that we only fit parameters for the non-negligible categories. For generality and clarity in notation, subsequent discussions are in terms of the full mark space $\mathcal{K}$.

The history-dependent discretized MkPP model is fit by solving a sequence of maximum likelihood problems. We assume that its parameters $\boldsymbol{\omega}_t$ admit piece-wise constant dynamics and are constant over consecutive windows of length $W$, where the log-likelihood of the $i^{\text{th}}$ window is denoted $\ell_i(\boldsymbol{\omega}_i)$. To encourage smoothly adapting parameter estimates, we use a forgetting factor mechanism [47] to adaptively weight the window log-likelihoods up to the $k^{\text{th}}$ window. For a forgetting factor $0 \leq \beta < 1$, the adaptively-weighted log-likelihood at window $k$ is thus defined as:

$$\ell_k^\beta(\boldsymbol{\omega}_k) := (1 - \beta)\sum_{i=1}^{k}\beta^{k-i}\ell_i(\boldsymbol{\omega}_k). \tag{3}$$

Parameter estimation is hence performed by solving the sequence of maximum likelihood problems:

$$\hat{\boldsymbol{\omega}}_k := \arg\max_{\boldsymbol{\omega}_k}\ \ell_k^\beta(\boldsymbol{\omega}_k), \quad k = 1, 2, \cdots, K. \tag{4}$$

To efficiently solve the sequence of problems in Eq (4) in an online fashion, we use the Adaptive Orthogonal Matching Pursuit (AdOMP) [48], an adaptive version of the Orthogonal Matching Pursuit (OMP) [49] [50]. The AdOMP, which fits an iteratively selected subset of model parameters, captures inherent sparsity of network interactions based on past ensemble activity [51–53] while also mitigating complexity. The implementation of AdOMP is addressed in detail in supporting information (S1 Appendix).

The sequence of maximum likelihood problems that must be solved to obtain the history-independent model takes a similar form as in Eq (4) under the same assumption of piece-wise

constant dynamics of $\boldsymbol{\mu}_t$. The sequence of maximum likelihood estimates

$$\hat{\boldsymbol{\mu}}_k = \arg \max_{\boldsymbol{\mu}_k} \ell_k^\beta(\boldsymbol{\mu}_k), \quad k = 1, 2, \cdots, K \tag{5}$$

are obtained by gradient descent, where the gradient of the history-independent log-likelihood can be computed directly. The procedure for computing the maximum-likelihood estimates of the history-independent model is detailed in supporting information (S1 Appendix).

## Statistical inference of higher-order coordination

Coordinated spiking can indicate relationships between components of a neuronal assembly and, potentially, effects of unobserved processes. However, independent neurons can also spike concurrently by chance, necessitating statistical inference to distinguish between excessive (or suppressed) and chance simultaneous spiking. To this end, we quantify the two alternatives as nested hypotheses and prove that an adaptive de-biased deviance test used for identifying significant Granger-causal influences [35] is applicable to our setting, thus establishing a precise statistical inference framework.

Here, we focused on characterizing the significance of $r^{\text{th}}$-order simultaneous spiking and have formulated the hypothesis test accordingly; however, similarly constructed null hypotheses can be used to test the significance of any set of simultaneous spiking events using the same inference procedure (see supporting information in S1 Appendix). For cogency, we focus on the statistical inference procedure for history-dependent MkPP models; differences for the history-independent model are addressed in supporting information (S1 Appendix). However, the complementary nature of the two models is summarized here. Theoretical results pertaining to the precise inference framework are summarized here, and comprehensively described in supporting information (S1 Appendix).

**Formulating nested hypotheses to test for $r^{\text{th}}$-order coordination**
The significance of $r$-wise simultaneous spiking for $r \geq 2$ is tested by considering the two alternatives:

$$
\begin{aligned}
H_0 \quad : \quad & r^{\text{th}}-\text{order simultaneous spikes occur as frequently as they would between} \\
& \text{independent units, given ensemble spiking history} \\
H_1 \quad : \quad & r^{\text{th}}-\text{order simultaneous spikes occur at a significantly different rate than} \\
& \text{they would between independent units, given ensemble spiking history}
\end{aligned}
\tag{6}
$$

A similar formulation was used in [39] to determine whether one mark occurs at a significantly different rate than expected. The likelihood of the mark was modeled as the product of marginal likelihoods and an additional multiplicative factor. Noting that the additional factor takes value 1 if the neurons are truly independent, the null hypothesis was quantified accordingly. Instead, we estimate a *reduced* model that assumes $r^{\text{th}}$-order interactions are chance occurrences by constraining the base rate parameters for each $r^{\text{th}}$-order mark.

The base rate parameter is decomposed as

$$\mu_k^{(m)} = \mu_{0,k}^{(m)} + \gamma_k^{(m)}, \tag{7}$$

where $\mu_{0,k}^{(m)}$ is the base rate under the null hypothesis and $\gamma_k^{(m)}$ is analogous to the additional multiplicative factor in [39] that captures potential exogenous effects after conditioning on ensemble spiking history. The reduced model thus constrains $\mu_{0,k}^{(m)} = \hat{\mu}_k^{(m)} - \hat{\gamma}_k^{(m)}$ for all

$r^{\text{th}}$-order marks, denoted $\mathcal{K}_r$, and solves the sequence of maximum likelihood problems $\hat{\boldsymbol{\omega}}_k^{(R)} := \arg\max_{\boldsymbol{\omega}_k^{(R)}} \ell_k^{\beta}(\boldsymbol{\omega}_k^{(R)})$.

The estimated exogenous factor at the $k^{\text{th}}$ window, $\hat{\gamma}_k^{(m)}$, is obtained as the average difference of the log-odds of the $m^{\text{th}}$ event under the null hypothesis, $u_{0,t}^{(m)}$, and under the alternative, $u_t^{(m)}$. That is,

$$\hat{\gamma}_k^{(m)} = \frac{1}{W} \sum_{t=(k-1)W+1}^{kW} \left( u_t^{(m)} - u_{0,t}^{(m)} \right). \tag{8}$$

**Precise statistical inference using deviance differences.**   The use of the deviance difference test statistic has been established in classical statistical methodology [54, 55] as a common procedure for likelihood ratio tests between two nested hypotheses. However, such a test is ill-suited to our setting due to the highly-dependent covariates and forgetting-factor mechanism in the data log-likelihood. These issues were addressed in a related context [35] for the inference of Granger-causal links by defining the *adaptive* de-biased deviance difference and characterizing its limiting distribution under presence and absence of Granger-causal links. We similarly utilize the adaptive de-biased deviance difference,

$$D_{k,\beta}^{(r)}\left( \hat{\boldsymbol{\omega}}_k^{(F)}, \hat{\boldsymbol{\omega}}_k^{(R)} \right) := \left( \frac{1+\beta}{1-\beta} \right) \left[ 2\left( \ell_k^{\beta}(\hat{\boldsymbol{\omega}}_k^{(F)}) - \ell_k^{\beta}(\hat{\boldsymbol{\omega}}_k^{(R)}) \right) - \left( \mathscr{B}_k^{(F)} - \mathscr{B}_k^{(R)} \right) \right], \tag{9}$$

as the test statistic, where $\mathscr{B}_k^{(F)}$ and $\mathscr{B}_k^{(R)}$ are the respective biases of the full and reduced models. The full and reduced model log-likelihoods can be efficiently computed online (see supporting information in S1 Appendix).

We precisely characterize the limiting behavior of the deviance difference in Eq (9) under both the null and alternative hypotheses, showing that as $\beta \to 1$:

i)  *if $r^{\text{th}}$-order coordination matches independent $r^{\text{th}}$-order interactions given ensemble spiking history, then $D_{k,\beta}^{(r)}(\hat{\boldsymbol{\omega}}_k^{(F)}, \hat{\boldsymbol{\omega}}_k^{(R)}) \xrightarrow{d} \chi^2(M^{(r)})$, i.e. chi-square, and*

ii) *if $r^{\text{th}}$-order coordination diverges from independent $r^{\text{th}}$-order interactions given ensemble spiking history, and assuming the base rate parameters of $r^{\text{th}}$-order interactions scale at least as $\mathcal{O}\left(\sqrt{\frac{1-\beta}{1+\beta}}\right)$, then $D_{k,\beta}^{(r)}(\hat{\boldsymbol{\omega}}_k^{(F)}, \hat{\boldsymbol{\omega}}_k^{(R)}) \xrightarrow{d} \chi^2(M^{(r)}, v_k^{(r)})$, i.e. non-central chi-square,*

where $v_k^{(r)}$ is the non-centrality parameter at time $k$ that depends only on the true parameters, and $M^{(r)} = |\mathcal{K}_r|$ is the difference in the cardinalities of the full and reduced support sets. Our theoretical results are comprehensively discussed in supporting information (S1 Appendix).

In order to fully characterize the limiting distribution of $D_{k,\beta}^{(r)}$ under $H_1$, we must estimate the non-centrality parameter for each window. Assuming it evolves smoothly in time, we use a state-space smoothing algorithm [35] to estimate $v_k^{(r)}$ from the observed $D_{k,\beta}^{(r)}$ values. Thus, in addition to identifying significant coordination, we also quantify the degree of significance using Youden's *J*-statistic

$$J_k^{(r)} := 1 - \alpha - F_{\chi^2(M^{(d)}, \hat{v}_k^{(r)})}\left( F_{\chi^2(M^{(d)})}^{-1}(1-\alpha) \right) \tag{10}$$

for significance level $\alpha$, where $F(\cdot)$ denotes the CDF. Values of $J_k^{(r)}$ close to 1 imply that the rejection of the null is a stronger indication of coordination than for smaller values of $J_k^{(r)}$. Thus, the *J*-statistic characterizes the test in terms of both type I and type II errors. By

convention, we take $J_k^{(r)} = 0$ when $H_0$ is not rejected at the $k^{\text{th}}$ window. Under the alternative, it is possible to observe either significant excess or suppressed coordination; this can be reflected in the $J$-statistic by incorporating the net exogenous effect on $r^{\text{th}}$-order coordination and using a signed $J$-statistic $J_k^{(r)} \cdot \text{sgn}(\sum_{m \in \mathcal{K}_r} \hat{\gamma}_k^{(m)})$. The full procedure for identifying significant $r^{\text{th}}$-order coordinated spiking is summarized by Algorithm 1.

**Algorithm 1** Dynamic inference of $r^{\text{th}}$-order spiking coordination

```
Input: {n_k^*}_{k=1}^K, r, β, α
Output: {J_k^(r)}_{k=1}^K, {v̂_k^(r)}_{k=1}^K, {D_{k,β}^(r)}_{k=1}^K
 1: 𝒦_r = {m ∈ 𝒦 : ∑_{c=1}^C m_c = r} and M^(r) = |𝒦_r|
 2: for k = 1 to K do
 3:    h_k = 0
 4:    Estimate ω̂_k^(F) using AdOMP; evaluate ℓ_k^β(ω̂_k^(F)) and ℬ_k^(F)
 5:    for m ∈ 𝒦_r do
 6:       Evaluate {u_t^(m)}_{t=(k-1)W+1}^{kW} and {u_{0,t}^(m)}_{t=(k-1)W+1}^{kW}
 7:       Set γ̂_k^(m) = 1/W ∑_{t=(k-1)W+1}^{kW} (u_t^(m) - u_{0,t}^(m)) and μ_{0,k}^(m) = μ̂_k^(m) - γ̂_k^(m)
 8:    end for
 9:    Estimate ω̂_k^(R) using AdOMP with constraint μ_k^(m) = μ_{0,k}^(m) for m ∈ 𝒦_r
10:    Evaluate ℓ_k^β(ω̂_k^(R)), ℬ_k^(R), and D_{k,β}^(r)(ω̂_k^(F), ω̂_k^(R))
11:    if F_{χ²(M^(r))}^{-1}(1-α) < D_{k,β}^(r)(ω̂_k^(F), ω̂_k^(R)) then
12:       h_k = sgn(∑_{m ∈ 𝒦_r} γ̂_k^(m))
13:    end if
14: end for
15: Estimate {v̂_k^(r)}_{k=1}^K via non-central χ² filtering/smoothing
16: J_k^(r) = h_k × (1 - α - F_{χ²(M^(r),v̂_k)}(F_{χ²(M^(r))}^{-1}(1-α)))
17: return {J_k^(r)}_{k=1}^K, {v̂_k^(r)}_{k=1}^K, {D_{k,β}^(r)}_{k=1}^K
```

In establishing the limiting behavior of $D_{k,\beta}^{(r)}$, we also proved a result of independent interest; namely, we generalized asymptotic properties of de-sparsified $\ell_1$-regularized estimates [56] to de-sparsified greedy estimates by showing that the de-sparsified AdOMP estimate behaves asymptotically like the maximum likelihood estimate. Crucially, this allows for the construction of confidence intervals around greedily estimated parameters, thus enabling precise statistical inference.

**Complementary characterizations of higher-order coordination by history-independent and history-dependent models.** The history-independent model is a special case of the history-dependent model; hence, the statistical inference procedure summarized by Algorithm 1 can be appropriately modified. This specialization is expounded in supporting information (S1 Appendix). However, the interpretation of statistically significant results obtained using history-independent analysis is distinct from but complementary to the interpretation of statistically significant results obtained using history-dependent model. Let the base rate parameter and exogenous effect for the history-independent model be denoted by $\mu_k$ and $\gamma_k$, respectively; and the same for the history-dependent model by $\mu_{k,\mathcal{H}}$ and $\gamma_{k,\mathcal{H}}$, with history-modulation parameter $\boldsymbol{\theta}_k$. Then, the reduced model constraints imply $\gamma_k = \gamma_{k,\mathcal{H}} + \bar{\boldsymbol{x}}_t^{\top} \boldsymbol{\theta}_k$. If the observed rate of higher-order events is equal to that of independent neurons, $\gamma_k = 0$; however, higher-order interactions may still be coordinated, i.e. $\gamma_{k,\mathcal{H}} = -\bar{\boldsymbol{x}}_t^{\top} \boldsymbol{\theta}_k \neq 0$. Conversely, the observed rate of higher-order events may differ from that of independent neurons, i.e. $\gamma_k \neq 0$. If $\gamma_{k,\mathcal{H}} = 0$, observed coordination can be attributed to the effects of ensemble history; otherwise, observed coordination was driven by an unobserved process. Thus, history-independent analysis reveals if the observed rate of simultaneous spiking events deviates from the expected

rate in a group in independent neurons, while history-dependent analysis reveals if the observed rate of simultaneous events cannot be attributed to endogenous network effects.

## Results

The proposed methods for analyzing higher-order spiking coordination were validated through simulations. First, we empirically verified our theoretical results, characterizing the limiting behavior of the adaptive debiased deviance difference on simulated spiking. Next, we demonstrated the utility of the analyses on simulated ensemble spiking data, showing that the ground-truth latent dynamics in higher-order coordinated spiking can be recovered with statistical confidence and with greater accuracy than existing single-trial metrics. Then, the proposed methods were applied to continuous multi-electrode recordings of human cortical assemblies during anesthesia and to rat cortical assemblies during sleep in order to infer latent dynamics in coordinated spiking during transitions between brain states.

Crucial hyperparameters that affect the estimation of MkPP models are the choice of bin size $\Delta$ (in physiological data); the window size $W$ over which parameters are assumed constant and the forgetting factor $\beta$, which define the effective integration window $N_{\text{eff}} = \mathcal{O}(\frac{W}{1-\beta})$ [57]; and the ensemble history integration window, i.e. the number of past samples spanned by the history coefficients $\boldsymbol{\theta}_k$. Justifications for hyperparameter selection are summarized in the following, while a detailed examination of their effects on MkPP model estimation is provided in supporting information (S2 Appendix).

### Empirical validation of the limiting behavior of deviance differences

We first validated the proposed statistical test for $r^{\text{th}}$-order coordinated spiking by empirically verifying the limiting distributions of the adaptive debiased deviance difference derived under the null and alternative hypotheses. We simulated and analyzed 50 realizations of a 5-neuron ensemble spiking process. Each realization was 4000 samples long with dynamics in $3^{\text{rd}}$-order spiking coordination. Namely, a step function was used to exogenously facilitate $3^{\text{rd}}$-order spiking during the second half of each realization.

The simulated spiking data were analyzed by applying Algorithm 1 to each realization; restricted models were only computed for $3^{\text{rd}}$-order spiking coordination. A threshold of $N_{thr}$ = 1 was used to pruned marked events that occurred no more than once per realization; this excluded $5^{\text{th}}$ order events from the estimated model. The window size over which parameters were assumed constant was set to $W = 10$ in order to enable stable estimation at each window while still allowing for fast changes. The forgetting factor was set to $\beta = 0.99$. For the purpose of validating the limiting distributions, a forgetting factor close to 1 was desirable; however, $\beta$ = 0.99 is also a practical choice for the forgetting factor that serves to illustrate the utility of the limiting distributions when analyzing physiological data. Indeed, the hyperparameter choices $(W, \beta) = (10, 0.99)$ were used in the applications to physiological data presented later. Hyperparameter choices for analysis are discussed in the context of two additional simulations.

The limiting distribution under the null hypothesis was validated by compiling adaptive debiased deviance differences that were computed during the first half of each realization and corresponded to small estimated non-centrality parameter values. Their distribution, depicted by the blue histogram in Fig 2, closely matched the theoretical distribution of deviance differences under the null hypothesis, i.e. a $\chi^2$ distribution with $M = 10$ degrees of freedom.

The limiting distribution under the alternative hypothesis was validated by compiling adaptive de-biased deviance differences that were computed during the second half of one particular realization and hence corresponded to similar estimated non-centrality parameter values.

### Theoretical vs. Empirical Deviance Difference Distribution

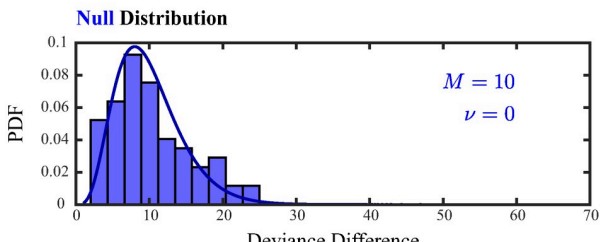

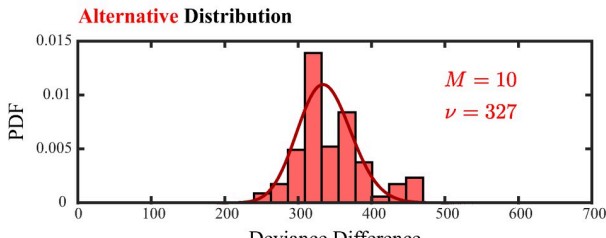

**Fig 2. Theoretical versus empirical distributions of the adaptive de-biased deviance difference under the null (blue) and alternative (red) hypotheses.** Empirical distributions of adaptive de-biased deviance differences were compiled from history-dependent analyses of 50 realizations of a discretized marked point process simulating the ensemble spiking of 5 neurons with exogenously induced $3^{rd}$-order spiking dynamics. The distributions were compared to the probability distribution functions (PDF) of a $\chi^2$ distribution with $M = 10$ degrees of freedom (left), and a non-central $\chi^2$ distribution with $M = 10$ degrees of freedom and non-centrality parameter $\nu = 327$ (right), corresponding respectively to the limiting distribution under the null and alternative hypotheses.

Their distribution, depicted by the red histogram in Fig 2, closely matched the theoretical distribution of deviance differences under the alternative hypothesis, a non-central $\chi^2$ distribution with $M = 10$ degrees of freedom and non-centrality parameter $\nu = 327$. The non-centrality parameter of the theoretical distribution was determined by computing the median of the aforementioned estimates.

## Simulated ensemble spiking: Example 1

We next demonstrated the utility of the proposed methods in application to two sets of simulated ensemble spiking data with dynamic latent higher-order coordination. An additional simulation addressing the utility of the proposed methods in analyzing large neuronal assemblies is included in supporting information (S2 Appendix). In the first example, spiking activity of 5 neurons was simulated for 16000 samples. Spiking activity (Fig 3A) included $3^{rd}$-order events exogenously facilitated and suppressed in alternation by a square wave and $4^{th}$-order events induced through endogenous effects of ensemble spiking history throughout the simulated duration. The latent spiking coordination was evident when visualizing the sums of all $r^{th}$-order events in Fig 3B. Four epochs of the simulation were defined by the periods of $3^{rd}$-order facilitation and suppression, shown in Fig 3C, and are indicated by vertical dashed lines common across all panels.

History-independent and history-dependent analyses of higher-order coordination were applied to the simulated spiking data using the same set of hyperparameters. A conservative threshold of $N_{thr} = 1$ was chosen to prune marked events that occurred unreliably over the simulated duration. The window size over which parameters were assumed constant was set to $W = 10$ in order to enable stable estimation at each window while still allowing for fast changes. Fixing $W$, several candidate values for the forgetting factor were considered to obtain the most appropriate effective integration window, $N_{eff} = \mathcal{O}(\frac{W}{1-\beta})$ [57], and was set to $\beta = 0.975$. For simulations the best choice of $\beta$ corresponded to $N_{eff} \approx \frac{\tau}{10}$, where $\tau$ denotes the duration of the shortest latent state; this heuristic is validated in supporting information (S2 Appendix). For the present simulation, $\tau = 4000$, the half-period of the square wave that defined alternating states of $3^{rd}$-order facilitation and suppression; hence, $N_{eff} = 400$. Statistical tests were performed at level $\alpha = 0.01$.

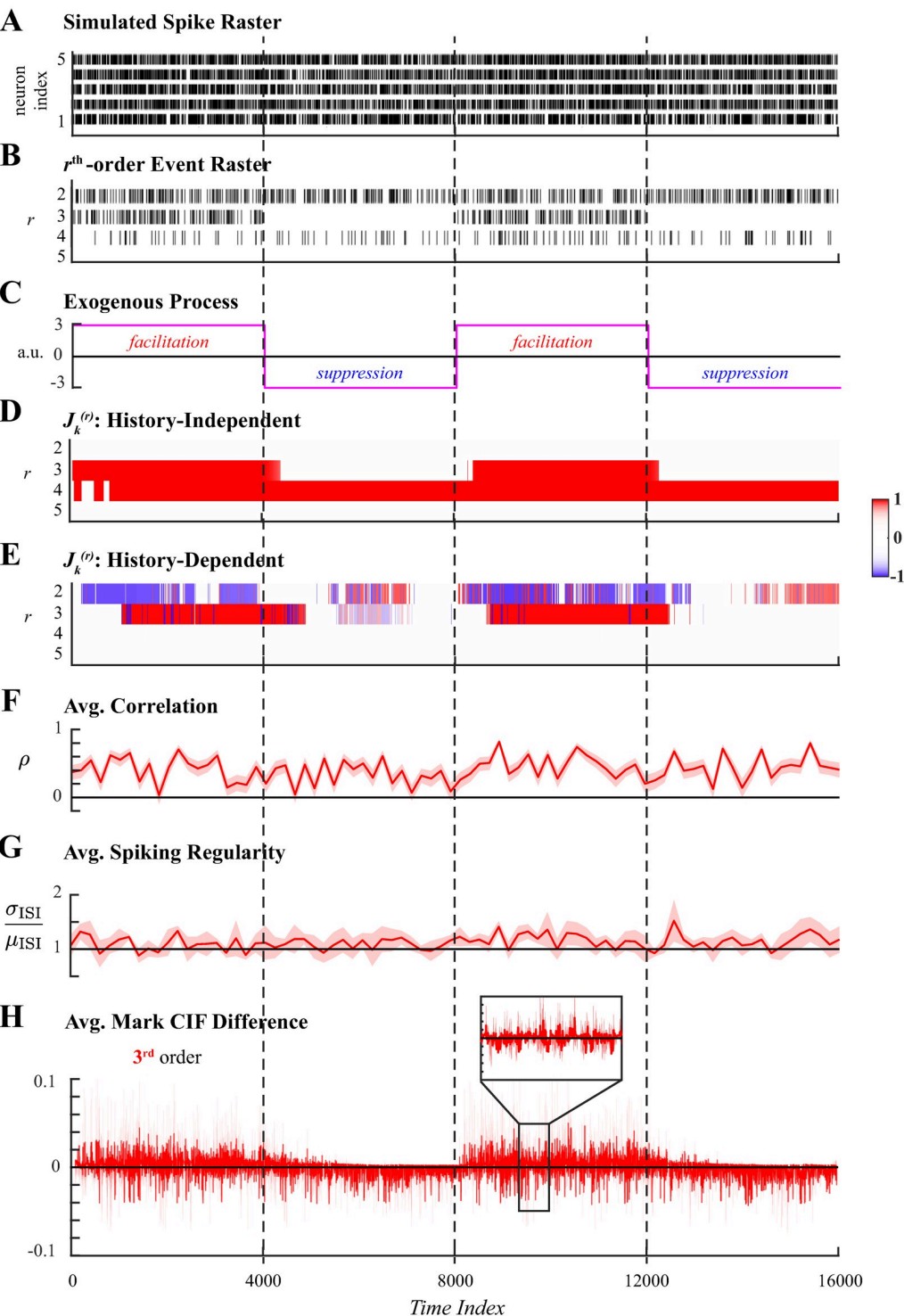

**Fig 3. Analysis of ensemble spiking with $3^{rd}$-order coordination induced exogenously by a square wave. A**. Simulated ensemble spiking of five neurons. **B**. Sum of the $r^{th}$-order simultaneous spiking events for $r$ = 2, 3, 4, 5. **C**. Spiking coordination varies across 4 epochs defined by the exogenous process that alternatingly facilitated and suppressed $3^{rd}$-order events. These are demarcated by vertical dashed lines. **D**. Significant $r^{th}$-order coordination neglecting ensemble history. **E**. Significant $r^{th}$-order coordination based on history-dependent ensemble spiking model. Statistical testing in **D–E** performed at level $\alpha$ = 0.01. **F**. Average Pearson correlation with 95% confidence interval. **G**. Average spiking regularity: coefficient of variation ±2 SEM. **H**. Average mark CIF differences of $3^{rd}$-order (green) spiking interactions ±2 SEM.

History-independent inference of higher-order synchrony (Fig 3D) accurately character-ized the periods of facilitated $3^{\text{rd}}$-order coordinated spiking and correctly assessed the rates of $4^{\text{th}}$-order spiking events to be significantly higher than expected amongst independent neu-rons, indicated by $J$-statistic values close to +1. However, $3^{\text{rd}}$-order suppression was not detected as a statistically significant, likely reflecting that the expected rate of $3^{\text{rd}}$-order was low to begin with and therefore difficult to distinguish. In complement, history-dependent infer-ence of higher-order coordination (Fig 3E) correctly attributed $4^{\text{th}}$-order spiking, which occurred at a statistically significantly high rate, to endogenous network effects captured by ensemble spiking history regressors while detecting that $3^{\text{rd}}$-order spiking was exogenously facilitated.

For comparison, three single-trial measures of coordinated spiking were utilized. The first is the average Pearson correlation between smoothed spiking responses. The second is the spiking regularity, quantified by the average coefficient of variation (ratio of the standard devi-ation to the mean inter-spike interval) [58]. A ratio close to 1 indicates Poisson spiking statis-tics; larger ratios indicate greater variability due to self-exciting dynamics while smaller ratios indicate regularity in spiking (i.e. globally coordinated spiking). Both measures were computed over non-overlapping windows of 200 samples to track dynamics, which while not identical are of a similar order of magnitude as $N_{\text{eff}}$. The third measure is the average difference between $r^{\text{th}}$-order mark CIFs and probabilities of $r^{\text{th}}$-order independent interactions, generalizing the measure employed in [38] to higher-order simultaneous spiking. Other existing model-based analyses require multiple trial repetitions and were thus unsuited to the single-trial simulation setting.

In application to simulated ensemble spiking data, the three control measures were unable to capture the latent dynamics in spiking coordination. Significant pairwise correlations (Fig 3F) were detected throughout the simulated duration, indicating only that several pairs of neu-rons were spiking concurrently, but were insensitive to the changes between facilitative and suppressive states of the exogenous process. Similarly, the spiking variability measure (Fig 3G) indicated Poisson-like spiking statistics throughout the simulation without reflecting any latent dynamics. The average mark CIF differences of $3^{\text{rd}}$-order events (Fig 3H) weakly reflected the dynamics of the exogenous process, but closer inspection (Fig 3H inset) reveals the oscillatory nature and wide confidence intervals of this sample-by-sample measure which pose challenges in interpreting the analysis.

## Simulated ensemble spiking: Example 2

The second simulated example utilized an autoregressive process instead of a square wave to induce exogenous $3^{\text{rd}}$-order coordinated spiking in a 5-neuron assembly. Ensemble spiking was simulated for 12000 samples (Fig 4A) with $3^{\text{rd}}$-order events exogenously induced by one realization of an autoregressive process. Additionally, $4^{\text{th}}$-order events were induced through endogenous effects for the first and last 4000-samples periods of the simulated duration, but occurred with chance-level probability otherwise. The sums of all $r^{\text{th}}$-order events (Fig 4B) reflected the latent spiking coordination. Coordinated $3^{\text{rd}}$-order spiking was most evidently facilitated during an interval when the exogenous variable had value greater than 2 (Fig 4C); the interval is indicated by vertical dashed lines common across all panels.

Both history-independent and history-dependent analyses were applied to the second simu-lated spiking data set using the same hyperparameters. The mark space was again pruned to include only events that occurred more than $N_{thr} = 1$ times; parameters were assumed constant over windows of $W = 10$ samples; and statistical tests were performed at level $\alpha = 0.01$. The

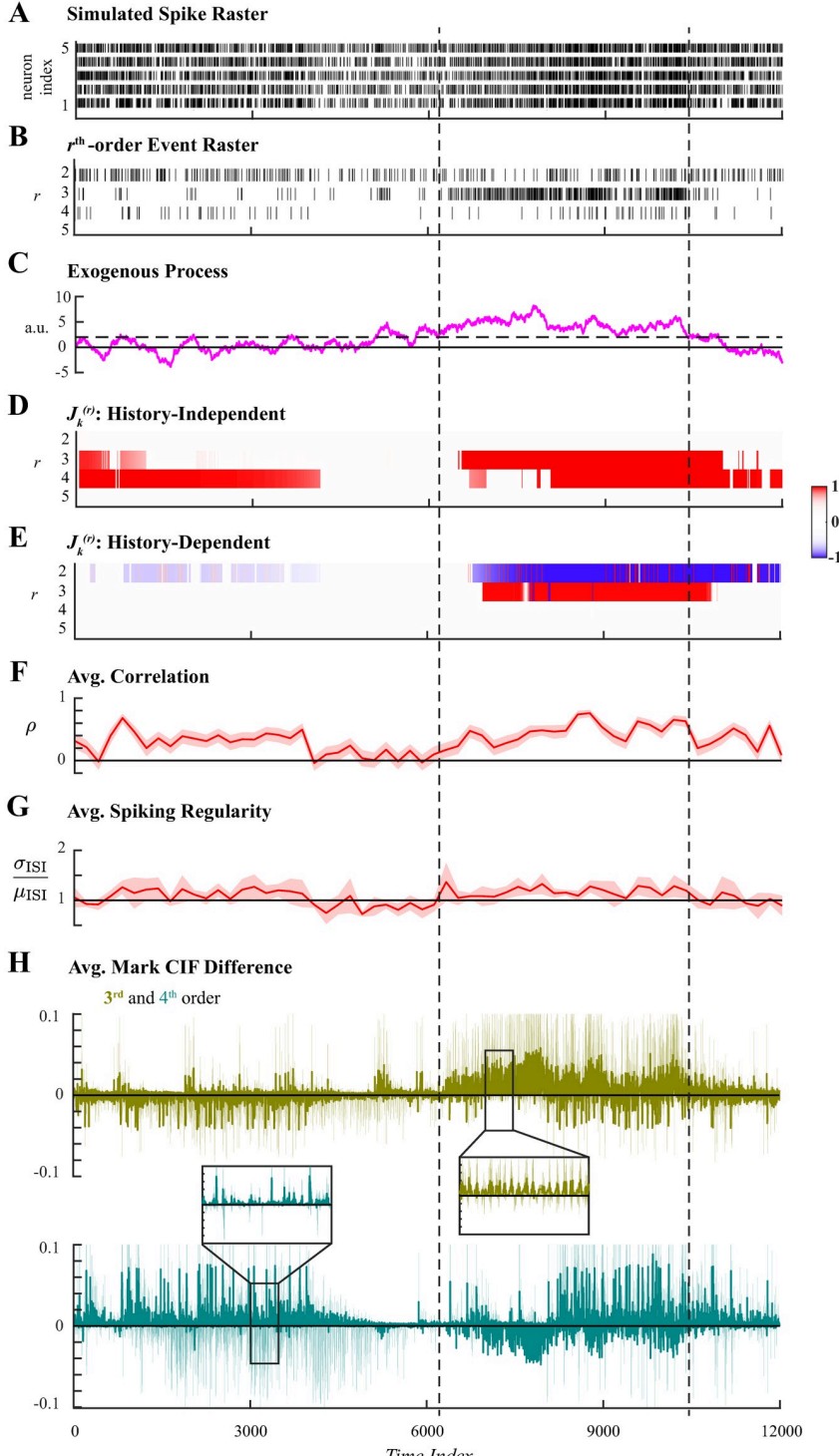

**Fig 4. Analysis of ensemble spiking with $3^{rd}$-order coordination induced exogenously by an autoregressive process. A**. Simulated ensemble spiking of five neurons. **B**. Sum of the $r^{th}$-order simultaneous spiking events for $r = 2$, 3, 4, 5. **C**. An autoregressive process was used to exogenously induce $3^{rd}$-order spiking coordination. The effect was most evident when the exogenous variable had value larger than 2 over an interval is demarcated by vertical dashed lines. **D**. Significant $r^{th}$-order coordination neglecting ensemble history. **E**. Significant $r^{th}$-order coordination based on history-dependent ensemble spiking model. Statistical testing in **D–E** performed at level $\alpha = 0.01$. **F**. Average Pearson correlation with 95% confidence interval. **G**. Average spiking regularity: coefficient of variation ±2 SEM. **H**. Average mark CIF differences of $3^{rd}$- (green) and $4^{th}$-order (teal) spiking interactions ±2 SEM.

forgetting factor was set to $\beta = 0.95$, which corresponded to an effective integration window $N_{\text{eff}} = 200$. We noted that the exogenous autoregressive process most persistently facilitated $3^{\text{rd}}$-order coordinated spiking for a duration of $\sim 4000$ samples; and within that duration, two subintervals of $\sim 2000$ samples separated at time index $\sim 8000$ can be discerned upon visual inspection (Fig 4C). Hence, taking $\tau = 2000$, the choice of $\beta = 0.95$ satisfies the criterion that $N_{\text{eff}} \approx \frac{\tau}{10}$.

The history-independent analysis of higher-order coordination correctly detected statistically significant $3^{\text{rd}}$-order coordination during the interval in which the exogenous variable was greater than 2 and $4^{\text{th}}$-order coordination when they were induced by ensemble spiking history (Fig 4D). The history-dependent analysis also correctly identified the exogenous facilitation of $3^{\text{rd}}$-order events while attributing $4^{\text{th}}$-order coordination to endogenous effects (Fig 4E). For comparison, the average Pearson correlation, average spiking regularity, and average mark CIF differences were computed in identical fashion as for the first simulation. The average Pearson correlation exhibited indicated significant pairwise correlations concurrently with both exogenously induced $3^{\text{rd}}$-order events and endogenously induced $4^{\text{th}}$-order events (Fig 4F); however, these two facets of latent higher-order coordination could not be disambiguated. In contrast, the average spiking regularity did not exhibit any dynamics; Poisson-like spiking statistics were indicated throughout the simulated duration (Fig 4G). The average mark CIF differences for $3^{\text{rd}}$- and $4^{\text{th}}$-order marks both weakly indicated the latent higher-order coordination (Fig 4H). In addition to previously issues concerning large confidence intervals and oscillatory nature, deviant average mark CIF differences for $3^{\text{rd}}$- and $4^{\text{th}}$-order events appear identical despite being induced in different manners. This illustrates that the average mark CIF differences only indicate when rates of $r^{\text{th}}$-order events deviate from the expected rate and cannot further address latent structure.

## Ensemble spiking in anesthetized humans

In the first application to recorded spiking data, we analyzed microelectrode recordings of human cortical neurons during the transition into propofol-induced general anesthesia. Commonly used in surgical procedures, general anesthesia is a drug-induced neurophysiological state of sedation and unconsciousness. In a study of the transition into unconsciousness, simultaneous recordings of single-neurons, LFP, and electrocorticograms were acquired to analyze changes to neural activity and functional connectivity over multiple spatial scales (full details of the experimental procedure are described in [59]). To complement previous analysis of pairwise spiking correlations, we employed the proposed methods for characterizing higher-order coordinated spiking.

Spiking data from the microelectrode recordings of one subject were analyzed, focusing specifically on the 8 neurons with the highest average firing rate over the 1000 second recording. Multi-unit spike recordings were originally oversampled at 1kHz, but downsampled by a factor of 50 to reduce computational complexity. Hence, the definition of simultaneous spiking in this analysis was taken to be the occurrence of spiking events across multiple neurons within at most 50ms of each other. This bin size selection was verified to minimize the coassignment of multiple spikes to the same bin in each of the 8 neurons' spike trains. Ensemble spiking activity is shown in Fig 5A, aligned to the loss of consciousness (LOC) at 0s when propofol was first administered; the effect was evident from the rapid decrease in spiking. Spiking activity recovered and after 250s propofol was administered again. In order to analyze higher-order coordination with the proposed methods, the mark space of $C^* = 2^8 - 1$ possible simultaneous spiking events, $\mathcal{K}$, was pruned to the set of reliable interactions $\bar{\mathcal{K}}$ that occurred more

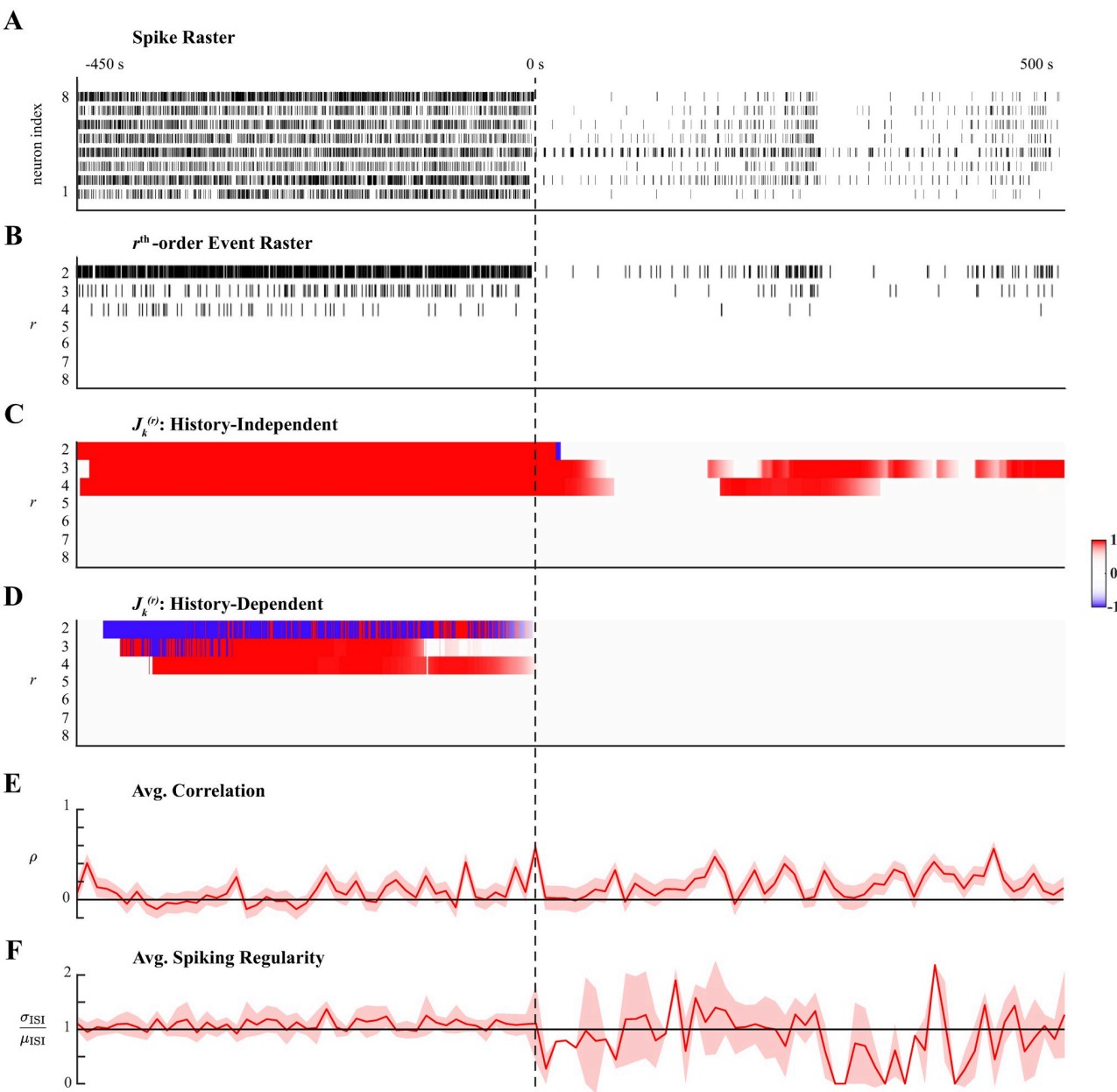

**Fig 5. Higher-order spiking coordination analysis of human cortical neurons during anesthesia. A.** Ensemble spiking of 8 neurons aligned to loss of consciousness (LOC) at 0s induced by administering propofol. A second administration of propofol occurred at $\sim$ 250s. **B.** Sum of the $r^{\text{th}}$-order simultaneous spiking events for $r = 2, \ldots, 8$. **C.** Significant $r^{\text{th}}$-order coordination neglecting ensemble history. **D.** Significant $r^{\text{th}}$-order coordination based on history-dependent ensemble spiking model. Statistical testing in **C–D** performed at level $\alpha = 0.01$. **E.** Average Pearson correlation with 95% confidence interval. **F.** Average spiking regularity: coefficient of variation ±2 SEM.

than $N_{thr} = 15$ times; that is, simultaneous spiking events with average rates less than 0.015Hz were treated as negligible. The cardinality of the set of reliable interactions, defined with a conservative threshold, was $|\bar{\mathcal{K}}| \approx 0.16 \cdot C^*$. The sums of all $r^{\text{th}}$-order events (Fig 5B) show that up to $4^{\text{th}}$-order coordinated spiking occurred reliably, though less frequently after LOC.

History-independent and history-dependent analyses were performed using the same hyperparameters. The window over which parameters were assumed constant was set to

$W = 10$. The forgetting factor, $\beta = 0.99$, was selected so that $N_{\text{eff}} \approx \frac{\tau}{5}$; here, we used $\tau = 5000$, the approximate number of samples between the two administrations of propofol. For applications to recorded data, choosing $\beta$ such that $N_{\text{eff}} \approx \frac{\tau}{10}$ (as was done for simulated data) yielded inferred higher-order coordination that was statistically weak (as quantified by $J$-statistics) and transient, resembling simulated examples where $N_{\text{eff}}$ was mismatched to the duration of latent states (supporting information in S1 Appendix). We speculate that a shorter effective integration was appropriate in simulations because the assemblies were comprised of 5 neurons with similar firing rates, which facilitated tracking latent dynamics. This contrasts with the variability in firing rates that can be observed in Fig 5A. Finally, statistical inference was performed at level $\alpha = 0.01$.

Applying history-independent higher-order coordination analysis revealed sustained significantly high rates of $2^{\text{nd}}$-, $3^{\text{rd}}$-, and $4^{\text{th}}$-order events prior to LOC (Fig 5C). Moreover, conditioning on ensemble spiking history indicated that $3^{\text{rd}}$- and $4^{\text{th}}$-order events were exogenously facilitated, while $2^{\text{nd}}$-order events were exogenously suppressed (Fig 5D). This latent structure was disrupted immediately following LOC; as spiking activity diminished, no higher-order coordination was detected. However, as spiking activity recovered, $3^{\text{rd}}$- and $4^{\text{th}}$-order events (but not $2^{\text{nd}}$-order events) occurred at significantly high rates. As the second administration of propofol again diminished spiking activity, the rate of $4^{\text{th}}$-order events became insignificantly different from the expect rate amongst independent neurons and did not recover. However, transient $3^{\text{rd}}$-order spiking after the second administration continued that the history-independent analysis detected as statistically significant. Third-order spiking was sustained at a significantly high rate once ensemble spiking activity recovered. Notably, none of the higher-order coordinated spiking after LOC was exogenously induced.

The dynamics in higher-order spiking coordination described by the proposed methods were poorly reflected by the average Pearson correlation and average spiking regularity. Both measures were computed over windows of 200 samples in order to track changes during the transition into anesthesia. Average correlations seemed to be significantly greater than zero for longer intervals after LOC than during consciousness, but trends in the average correlation were difficult to distinguish (Fig 5E). The average spiking regularity measure indicated Poisson-like spiking statistics throughout, contrasting the dynamics of higher-order coordination described by the proposed analyses. Average spiking regularity was ill-suited to analyzing dynamics after LOC due to the reduced spiking activity; this was reflected by abrupt changes and wide confidence intervals (Fig 5E).

In summary, history-independent and history-dependent analyses of ensemble spiking during the transition into anesthesia revealed the rapid onset of differences in latent higher-order coordination that distinguished consciousness from anesthesia. Specifically, comparisons between the history-independent and history-dependent results suggest that exogenous influences on the higher-order interactions of small neuronal assemblies during consciousness are disrupted during anesthesia. These results are corroborated by previous analyses of these data [59] that indicated a rapid state change in which local network interactions were preserve but spatially distant network interactions were disrupted during anesthesia. Previous studies have shown that propofol acts by enhancing GABAergic circuits whose recurrent dynamics contribute to inducing synchronized slow-wave oscillatory activity [59–63]. Ensemble spiking history regressors likely accounted for these recurrent dynamics in the history-dependent model so that no exogenous effects were detected.

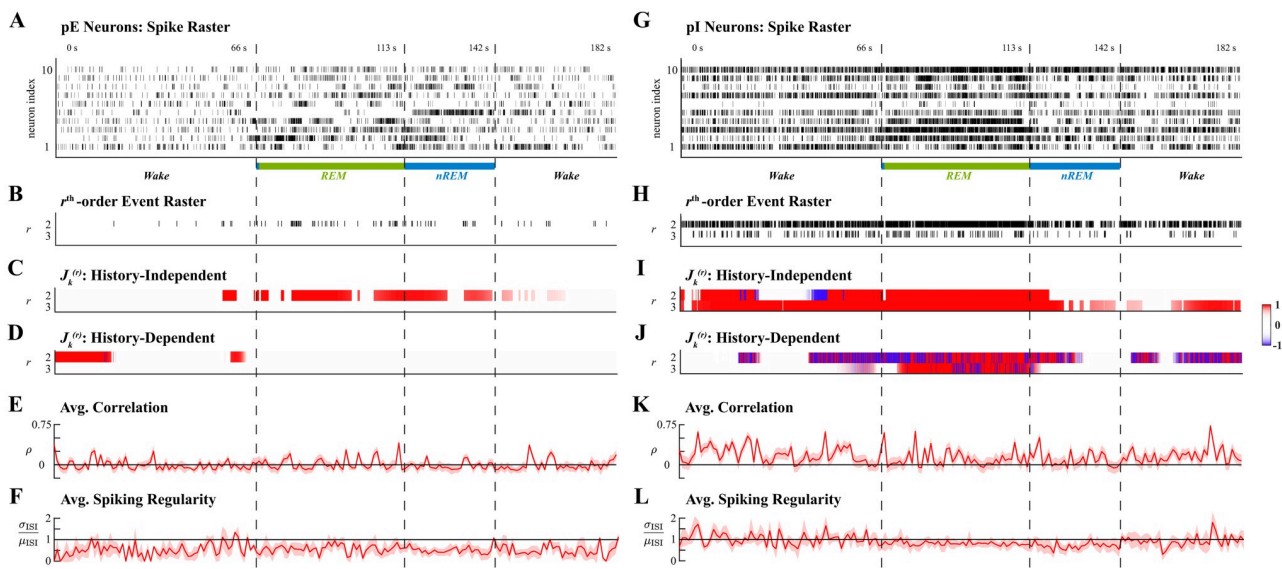

**Fig 6. Higher-order spiking coordination analysis of excitatory (pE) and inhibitory (pI) rat cortical neurons during one sleep cycle.** Left and right columns show analyses of pE and pI neurons, respectively. **A**. Ensemble spiking of 10 pE neurons. **B**. Sum of the $r^{th}$-order simultaneous spiking events for $r = 2, 3$. **C**. Significant $r^{th}$-order coordination neglecting ensemble history. **D**. Significant $r^{th}$-order coordination based on history-dependent ensemble spiking model. **E**. Average Pearson correlation with 95% confidence interval. **F**. Average spiking regularity: coefficient of variation ±2 SEM. **G**. Ensemble spiking of 10 pI neurons. **H**. Sum of the $r^{th}$-order simultaneous spiking events for $r = 2, 3$. **I**. Significant $r^{th}$-order coordination neglecting ensemble history. **J**. Significant $r^{th}$-order coordination based on history-dependent ensemble spiking model. **K**. Average Pearson correlation with 95% confidence interval. **L**. Average spiking regularity: coefficient of variation ±2 SEM. Statistical testing in **C–D, I–J** performed at level $\alpha = 0.01$.

## Ensemble spiking in sleeping rats

We additionally analyzed ensemble spiking data recorded from rat cortical neurons during sleep. Sleep consists of cyclical transitions between brain states that maintain homeostatic neural activity distinct from waking states; however, both the purpose and mechanisms of these transitions remain unclear. We analyzed large-scale spike recordings from frontal and motor cortices during sleep obtained to study the effects of different sleep stages on the firing rate dynamics of putatively excitatory (pE) pyramidal neurons and putatively inhibitory (pI) interneurons [64, 65]. By examining neuronal activity recorded during several instances of rapid eye movement (REM), non-REM (nREM), and microarousal states over multiple sleep cycles, the study sought to address homeostatic effects of sleep. Instead, we sought to use the proposed analyses of higher-order spiking coordination to study the dynamics during transitions into sleep and between REM and nREM states in one sleep cycle.

We analyzed spiking data during one 182s long sleep cycle from one animal in which at least 10 pE and pI neurons were identified, selecting the 10 neurons of each class with the highest average firing rate. Recordings were originally oversampled at 20kHz, but downsampled to 200Hz to reduce computational complexity. Simultaneous spiking in this analysis hence equated to the occurrence of spiking events across multiple neurons within at most 5ms of each other. This bin size selection was verified to minimize the coassignment of multiple spikes to the same bin in each of the 10 neurons' spike trains in both populations. Ensemble spiking activity of pE and pI neurons were analyzed separately; the activity of each population during the sleep cycle is shown in Fig 6A and 6G, respectively, annotated by arousal states. The cycle analyzed consisted of a 66s wake-period, a transient 1s nREM period, a 46s REM period, a 29s nREM period, and finally a 40s wake-period.

For tractable analysis, the mark spaces of $C^* = 2^{10} - 1$ possible simultaneous spiking events, $\mathcal{K}$, of both populations were pruned to the set of reliable interactions $\bar{\mathcal{K}}$ that occurred more than $N_{thr} = 10$ times; that is, simultaneous spiking events with average rates less than 0.055Hz were treated as negligible. The cardinality of the set of reliable interactions amongst pE neurons was $|\bar{\mathcal{K}}_{pE}| \approx 0.017 \cdot C^*$ and amongst pI neurons was $|\bar{\mathcal{K}}_{pI}| \approx 0.058 \cdot C^*$. The sums of all $r^{\text{th}}$-order events (Fig 6B and 6H) show that up to $3^{\text{rd}}$-order coordinated spiking occurred reliably amongst pI neurons while only up to $2^{\text{nd}}$-order interactions occurred reliably amongst pE neurons. The same effective integration windows were used for history-independent and history-dependent analyses of both neuronal populations; with $W = 10$, the forgetting factor $\beta = 0.99$ so that $N_{\text{eff}} = \frac{\tau}{5}$, where $\tau \approx 5000$ was the duration of the second nREM interval. Statistical inference was performed at level $\alpha = 0.01$.

Applying the history-independent and history-dependent analyses of higher-order coordination to the ensemble spiking of pE neurons in concert identified intervals of significantly higher rates of $2^{\text{nd}}$-order events that could be attributed to effects of ensemble spiking history (Fig 6C and 6D). Most of the detected intervals were not sustained during either REM or nREM sleep; rather, they were aligned to the transitions between states. However, pI neurons exhibited more structured higher-order coordination. History-independent analysis of pI neurons revealed that $2^{\text{nd}}$-order events had significantly higher rates during two intervals; the first was during the wake-period, and the second started at the end of the first wake-period and ending at the transition from REM to nREM sleep (Fig 6I). While during the first of these intervals the facilitation of $2^{\text{nd}}$-order events could largely be attributed to ensemble history effects, there was a shift in the exogenous effects on $2^{\text{nd}}$-order during the second interval (Fig 6J). That is, after the transition from the wake state to REM sleep, the exogenous suppression of $2^{\text{nd}}$-order events gradually shifted to exogenous facilitation by the middle of the REM period that persisted into the nREM period. Exogenous $2^{\text{nd}}$-order coordination was no longer detected strongly after the first half of the nREM period, but exogenous suppression emerged again in the second wake-period.

In addition to dynamics in $2^{\text{nd}}$-order coordination, pI neurons also exhibited significant $3^{\text{rd}}$-order coordination. The rate of $3^{\text{rd}}$-order events was significantly high during the first wake period and REM sleep; though significantly higher at the start of nREM sleep, only statistically weak and transiently high rates were detected during the middle and end of nREM sleep. However, in the second wake period, the rate of $3^{\text{rd}}$-order coordinated events again became significantly high (Fig 6I). Notably, the high rate of $3^{\text{rd}}$-order events during REM was distinctive because it was exogenously facilitated, whereas $3^{\text{rd}}$-order events during other periods occurred at significantly high rates because of endogenous effects (Fig 6J).

In contrast to the proposed analyses, neither the average Pearson correlation nor average spiking regularity, computed over windows of 200 samples, reflected similar latent dynamics of higher-order coordination amongst pE or pI neurons. For pE neurons, pairwise correlations were close to 0 for much of the sleep cycle with the exception of a few windows (Fig 6E). However, the spiking regularity was significantly less than 1 for much of the sleep cycle (Fig 6F); the implication of globally coordinated ensemble spiking is at odds with the absence of reliably occurring higher-order spiking events amongst pE neurons. For pI neurons, the average correlation was significantly higher than 0 during the first and second wake periods, mirroring the significantly high rates of higher-order events during these intervals; however, excepting a few windows, the average correlation did not significantly differ from 0 during REM sleep (Fig 6K), presenting an inconsistency with the rates of higher-order events. Meanwhile, the average spiking regularity did not differ significantly from 1 for most of the sleep cycle, indicating

Poisson-like spiking activity (Fig 6L); this contrasts sharply from the reliable occurrence of higher-order events.

Summarily, applying the history-independent and history-dependent analyses of higher-order spiking coordination revealed distinctive latent dynamics amongst pE and pI neurons during the same sleep cycle. Intervals of significant $2^{nd}$-order spiking coordination amongst pE neurons were attributable to the effects of ensemble spiking history and occurred around the transitions between arousal states rather than being sustained during the arousal states, possibly relating to a hypothesis that transition periods are themselves distinct states [66]. In contrast, $2^{nd}$- and $3^{rd}$-order spiking events amongst pI neurons were detected to be exogenously coordinated, especially during sleep states. The observed changes in higher-order coordination of pI neurons during REM sleep are consistent with previous results that have shown excitation of pI neuronal activity and coordination during REM sleep [67, 68]. Additionally, the detected exogenous influences on pI neurons may be explained by studies that have indicated signatures of REM sleep can be found in hippocampal neurons prior to cortical neurons [69, 70].

## Discussion and concluding remarks

### Relations to other models of coordinated spiking activity

The proposed algorithms integrate some notable functionalities of existing maximum entropy model variations with the GLM framework, and are tailored for the analysis of continuously acquired neuronal data. As Truccolo's comparisons in [24] suggest, GLMs account for temporal dynamics explicitly in modeling ensemble spiking, and thus are arguably more predictive than maximum entropy models. Within the context of the MkPP mGLM we utilized, temporal dynamics of neuronal spiking were modeled as relevant covariates in the estimation of ensemble spiking events. Such a model can be simplified to exclude spiking history, as demonstrated by the history-independent model; and can be expanded to model the influence of stimuli, as previously addressed for maximum entropy models [30].

Due to the large number of possible interactions, challenges in the tractability of synchrony analyses are inherent, particularly when modeling the effects of relevant covariates. Incorporating the emphasis on reliable interactions, as proposed in [31], model complexity may be managed in a data-driven fashion. The proposed adaptive greedy filtering algorithm for sparse model estimation ensures only the salient effects of covariates are captured. The adaptive filtering algorithm also characterizes dynamics in network correlational structure, analogous to Bayesian state space filtering algorithms [28, 29], and is thus applicable in the analysis of non-stationary neuronal processes. In lieu of constructing credible intervals around the aforementioned Bayesian estimates, we utilize a statistical test for which the test statistic's limiting distribution is precisely characterized. Unlike existing analyses, the proposed statistical tests do not require repeated trials of data to detect coordinated spiking activity, and are thus suitable for the analysis of continuous recordings of ensemble neuronal spiking.

Extending previous results in high-dimensional statistics, we have shown in Theorem 1 that the elegant procedure of [56] for LASSO estimation may be adapted to de-sparsify OMP estimates, and that de-sparsified estimates are asymptotically normal. In reviewing the existing literature, we noted a paucity in work on variable selection algorithms concerning the construction of confidence intervals. The OMP has been shown to have similar consistency properties as LASSO regression under appropriate conditions [49, 50]; however, in settings with large quantities of data, the latter becomes intractable. The result established by Theorem 1 enables the construction of confidence intervals around OMP-estimated parameters in order

to provide analogous methods of statistical inference as LASSO for an algorithm suitable in settings with large data sets, addressing this gap in the high-dimensional statistics literature.

## Novel insights into coordinated network activity

The proposed modeling and statistical inference framework constitute a novel approach to studying coordinated neuronal spiking by enabling the adaptive analysis of continuously acquired or single-trial data. The ability to track dynamics and detect exogenous influences on ensemble spiking with statistical confidence provides a new approach to probing the neural mechanisms underlying transitions between and characteristics of arousal states.

Simulated data examples verified the recovery of underlying correlational structure in ensemble spiking. In particular, the simulation results emphasized the distinction that the proposed method makes between synchrony and coordination based on comparisons of the history-independent and history-dependent version of the analysis, respectively.

In applications to physiological data, we first analyzed ensemble spiking of human cortical neurons during the transition into anesthesia. Directly comparing our results to previous insights gained from the same data in [59], the proposed method was consistent in indicating the rapid onset of disrupted global connectivity but the preservation of local connectivity during anesthesia. Absent a ground truth, this comparison substantiated insights gained from applying the proposed methods to physiological data.

Next, we analyzed the ensemble spiking activity of rat cortical neurons during one sleep cycle. State transitions during sleep have typically been characterized in terms of multiband analysis of electrophysiological recordings [66, 67, 69, 70]; meanwhile, to the best of our knowledge, properties of neuronal spiking in different sleep states have been characterized non-parametrically (e.g. with correlations, mean activity, Fano Factor, etc.) [64, 67, 68]. Hence, spiking dynamics over a single sleep cycle appear not to be well-explored. The novel insights into spiking coordination during fast state transitions over the course of a single sleep cycle provided by the proposed methods serve to motivate future studies of the correlates of state transitions at a fine spatiotemporal resolution.

## Extensions

The proposed statistical inference framework was developed to test for significant coordination of $r^{th}$-order spiking events, and the presented results demonstrated its efficacy. Specifically, Theorem 2 characterized the limiting distributions for the adaptive de-biased deviance difference test statistic under both outcomes of a nested hypothesis test in which the null hypothesis restricted parameters to impose conditionally independent $r^{th}$-order spiking. However, a nested null hypothesis can, in principle, be constructed to impose different assumptions. An immediate extension of the proposed analysis could include spatial information, for example, so that a null hypothesis assumes $r^{th}$-order spiking amongst a spatially localized subset of a recorded neuronal assembly is conditionally independent. The proposed inference framework was hence established to readily extend to any nested hypothesis test in Corollary 2.2.

An important consequence of this corollary result is that it provides a theoretical foundation for adaptive Granger causality using greedy algorithms. Since the proposed methods utilize a multinomial extension of generalized linear models, Corollary 2.2 establishes the asymptotic result in [35] for greedy parameter estimates in the limiting case of a single-neuron model. Notably though, Corollary 2.2 also implies that a nested hypothesis test can be formulated to determine if exogenous signals, such as sensory stimuli or concurrent activity in other brain regions, have Granger-causal effects on a neuronal network or its subsets. Thus, the

methods proposed in the present study can be extended to investigate the local network effects of global neural dynamics.

## Supporting information

**S1 Appendix. Algorithms, derivations, and theoretical results.** In this appendix, we present supporting information regarding algorithm development and our theoretical results supporting the proposed statistical inference framework.
(PDF)

**S2 Appendix. Supporting simulations.** In this appendix, we present supporting simulations that address hyperparameter selection and the scalability of the proposed method to large neuronal assemblies.
(PDF)

## Acknowledgments

We thank Emery N. Brown and Patrick L. Purdon for providing neural data (published in [59]) recorded from human cortical assemblies during anesthesia.

## Author Contributions

**Conceptualization:** Shoutik Mukherjee, Behtash Babadi.

**Data curation:** Shoutik Mukherjee.

**Formal analysis:** Shoutik Mukherjee.

**Funding acquisition:** Behtash Babadi.

**Investigation:** Shoutik Mukherjee.

**Methodology:** Shoutik Mukherjee.

**Project administration:** Behtash Babadi.

**Software:** Shoutik Mukherjee.

**Supervision:** Behtash Babadi.

**Validation:** Shoutik Mukherjee.

**Visualization:** Shoutik Mukherjee.

**Writing – original draft:** Shoutik Mukherjee, Behtash Babadi.

**Writing – review & editing:** Shoutik Mukherjee, Behtash Babadi.

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
