## [Decision Letter · Decision Letter 0]

2 Jan 2024

Dear Prof. Babadi,

Thank you very much for submitting your manuscript "Adaptive modeling and inference of higher-order coordination in neuronal assemblies: a dynamic greedy estimation approach" for consideration at PLOS Computational Biology.

As with all papers reviewed by the journal, your manuscript was reviewed by members of the editorial board and by several independent reviewers. In light of the reviews (below this email), we would like to invite the resubmission of a significantly-revised version that takes into account the reviewers' comments.

In particular, the authors should ensure that the relationship between their method and previous, related algorithms is well described, to provide better context for the novel contribution made here; and that the mathematical details are complemented by a more intuitive description of their approach which will increase the impact of this manuscript by making it more accessible, particularly to experimental neuroscientists. To that end, it may even be worth moving some mathematical details to the supplementary information, if they feel that this could improve the readability of the paper.

We cannot make any decision about publication until we have seen the revised manuscript and your response to the reviewers' comments. Your revised manuscript is also likely to be sent to reviewers for further evaluation.

Sincerely,

Daniel Bush

Academic Editor

PLOS Computational Biology

Daniele Marinazzo

Section Editor

PLOS Computational Biology

In particular, the authors should ensure that the relationship between their method and previous, related algorithms is well described, to provide better context for the novel contribution made here; and that the mathematical details are complemented by a more intuitive description of their approach which will increase the impact of this manuscript by making it more accessible, particularly to experimental neuroscientists. To that end, it may even be worth moving some mathematical details to the supplementary information, if they feel that this could improve the readability of the paper.

Reviewer's Responses to Questions

**Comments to the Authors:**

Reviewer #1: The authors propose a novel method to detect significant high-order correlations in multi-neuron spike trains in the absence of trial structure. The methodology appears rigorous, although the mathematical details are quite dense and not possible for me to exhaustively verify. The scientific significance of these findings is not communicated clearly, and the paper should include more head-to-head comparisons with baseline models.

Below I comment on the elements of the manuscript laid out in the reviewer guidelines.

ORIGINALITY + INNOVATION. The method seems original, but is not directly compared to previous methods on simulated or real data in a head-to-head manner. The authors argue that previous approaches "require multiple repeated trials to capture dynamics in correlational structure." But it isn't immediately clear why this is true, particularly since one of the baseline methods they mention is a state space model -- most state space models can be fit on unstructured time series data. I think at minimum there needs to be a section where the authors write down equations of prior models and explain why those models require multiple trials. It would be even better if the authors formulate some naive extensions of these existing models to the trial-free data and show how those naive extensions fail, whereas their approach succeeds. Currently, the paper is not self-contained, making it difficult for a reader to digest without first reading a long and niche sub-literature.

IMPORTANCE TO RESEARCHERS. The authors cite a number prior works that have developed similar statistical methodologies. The first paragraph of the introduction also mentions some motivations for the analysis from experimental neuroscience, but these are not really developed. Insights derived from experimental analysis could also be communicated more clearly -- for example, the authors observe changes in correlation structure associated with loss of consciousness, but it is not clear whether this could be used for a clinically relevant purpose or if simpler methods would not be sufficient for this application.

SIGNIFICANT BIOLOGICAL OR METHODOLOGICAL INSIGHT. As explained above I think the biological insights are weak and speculative. But there are clearly methodological insights and innovations in this paper.

RIGOROUS METHODOLOGY. The methodology is complex and clearly the authors have put a lot of thought into it. They verify that the method behaves as expected on simulated data. On the other hand, there are elements of the method that appear to be ad hoc or heuristic. For example, the authors threshold a subset of the mark space with a parameter (N_thr) on page 6. It is clear why this is necessary for computational reasons, but it is less clear how users should set this value in practice, particularly since the number of thresholded marks will change as a function of the length of the recording. Does this thresholding introduce bias into the confidence intervals or statistical tests?

Similarly, the window length W and forgetting factor beta seem like very important parameters that need to be tuned. The authors seem to have an intuition for how to pick these parameters, but it isn't spelled out clearly for the reader. For instance, on page 13 say they fix W to an intermediate value to trade off between "stable estimation... while still allow for fast changes." How should one quantify stable estimation? In Figure 3E, there is often flickering between red and blue, which would seem to be unstable predictions?

After fixing W, the authors perform a sweep over the forgetting factor beta to achieve "the most appropriate value." What exactly is being optimized here? It seems like if these parameters are chosen incorrectly it will lead to a loss of statistical power, while on the other hand if they are optimized to create low p-values it would inflate type-I errors. I unfortunately cannot read supporting information S1 Appendix which seems important to these issues. Was it uploaded?

SUBSTANTIAL EVIDENCE FOR CONCLUSIONS. The point of this paper is to develop a statistical methodology and not to develop a scientific theory or test specific hypotheses (as far as I can tell). So this point does not seem particularly applicable. However, as mentioned above under "originality + innovation" I would like to see more clear discussions and comparisons with existing statistical methods. Ideally the authors should show how their method outperforms others head-to-head on simulated datasets.

Reviewer #2: This work proposes a well-substantiated methodology to assess higher-order coordination in neural spiking activity, as well as a framework for accurate statistical inference to rule out occurrences of random simultaneous spiking events. The methodology is validated in simulated settings and applied to neural spike train recordings of human and rat cortical assemblies. The experimental results are interesting, revealing the onset of different higher-order coordination patterns distinguishing consciousness from anesthesia in the human data, and sleep state transitions in the animal data. The paper is accurately written, though resulting rather difficult to follow for non-expert readers especially in the methodological description. A better placing in the context of the existing literature and description of the dependence on crucial parameters are suggested.

Specific comments:

1) The proposed method is based on statistical model representations, which are contrasted to more general model-free approaches only briefly in the introduction. Nevertheless, an emerging line of research for the model-free analysis of multiple spike trains, stemming from the methodological and applicative works listed below [R1-R4], seems overlooked in the present paper. These works, showing the feasibility and usefulness of multivariate model-free analysis of causal (history-dependent) interactions and synchronization in spike trains, should be acknowledged, and compared – at least at the level of discussion – with the statistical models on which the present contribution is grounded.

[R1] Shorten, D. P., et al. (2021). Estimating transfer entropy in continuous time between neural spike trains or other event-based data. PLoS computational biology, 17(4), e1008054.

[R2] Mijatovic, G., et al. (2021). An information-theoretic framework to measure the dynamic interaction between neural spike trains. IEEE Transactions on Biomedical Engineering, 68(12), 3471-3481.

[R3] Shorten, D. P. et al. (2022). Early lock-in of structured and specialised information flows during neural development. Elife, 11, e74651.

[R4] Mijatovic, G. et al. (2022). Measuring the rate of information exchange in point-process data with application to cardiovascular variability. Frontiers in Network Physiology, 1, 765332.

2) Since the mathematics behind the derivations is a bit heavy, it should be complemented by a more intuitive description of the features of the proposed methodology, where its properties are presented in layman terms; from this viewpoint, improving Fig. 1 and accompanying it with descriptive text would be useful.

3) Analysis parameters such as the data length, the bin width, and the length of the past window considered for history-dependent formulation of the method are crucial aspects in approaches to neural synchrony based on binning the temporal axis. To better understand the performance of the proposed methodology, its dependence on these parameters, as well as on their interplay, should be tested and discussed.

**Have the authors made all data and (if applicable) computational code underlying the findings in their manuscript fully available?**

Reviewer #1: Yes

Reviewer #2: Yes

PLOS authors have the option to publish the peer review history of their article (what does this mean?). If published, this will include your full peer review and any attached files.

Reviewer #1: No

Reviewer #2: No
---

## [Decision Letter · Decision Letter 1]

19 Mar 2024

Dear Prof. Babadi,

Thank you very much for submitting your manuscript "Adaptive modeling and inference of higher-order coordination in neuronal assemblies: a dynamic greedy estimation approach" for consideration at PLOS Computational Biology. As with all papers reviewed by the journal, your manuscript was reviewed by members of the editorial board and by several independent reviewers. The reviewers appreciated the attention to an important topic. Based on the reviews, we are likely to accept this manuscript for publication, providing that you modify the manuscript according to the review recommendations.

Unfortunately, we have had to recruit an additional reviewer at this stage because one of the original referees was unable to assess the revised manuscript. Both this new reviewer and myself feel that a little more work is needed to demonstrate the practical utility of this new method - first, by making some direct comparisons with established and widely used methods based on PCA or ICA, for example; and second, demonstrating that the method can be applied to larger scale data sets consisting of tens or hundreds of simultaneously recorded neurons, as are increasingly common in contemporary neuroscience.

Sincerely,

Daniel Bush

Academic Editor

PLOS Computational Biology

Daniele Marinazzo

Section Editor

PLOS Computational Biology

Reviewer's Responses to Questions

**Comments to the Authors:**

Reviewer #2: The paper has been revised addressing all the concerns raised. In this reviewer opinion the paper deserves acceptance.

Reviewer #3: In this manuscript, the authors introduce a methodological framework that employs a discretized marked point process model with adaptive estimation techniques and statistical inference procedures to analyze coordinate ensemble activity. While the sophistication of the method allows for a nuanced representation of neural interactions, its complexity raises concerns regarding overfitting, generalizability, and statistical power. Furthermore, the mathematical presentation is densely packed and lacks intuitive explanations, which may limit accessibility to a broader audience. The framework demands a significant time investment for comprehension, potentially obscuring statistical limitations and impacting interpretability. Although complexity may be necessary for the proposed goals, the study does not convincingly demonstrate this necessity.

The necessity of such a complex framework is questioned by comparing it to existing, simpler methods known for detecting high-order correlations and co-activity patterns, such as those based on PCA or ICA (Chapin and Nicolelis 1999, Laubach et al. 1999, Peyrache et al. 2009, Lopes-dos-Santos et al. 2013). These approaches offer better interpretability and are widely used despite assumptions of linearity, non-stationarity, and being memoryless. The authors point out that their method does not suffer from these limitations but have not shown through simulations that it would outperform these simpler methods in practice due to these limitations. This comparison is crucial for justifying the new approach's complexity and provides a benchmark for assessing the proposed method's incremental benefits, highlighting the importance of balancing sophistication with accessibility.

Furthermore, the authors present their methodological advancements through the analysis of two simulated datasets, which, while illustrative of the method's potential, limit the demonstration of its versatility and robustness across a broader spectrum of neural coordination patterns. The simulations do not explore variations in coordination dynamics such as differing time scales, the size of neuronal assemblies, or the presence of multiple coordinating assemblies. Moreover, the choice to simulate the activity of only five neurons raises concerns about the method's scalability and its ability to maintain statistical power when faced with the increased dimensionality typical of larger neuronal ensembles. This limitation gives the impression that the method may not be well-suited to handle the complexity of real-world neural data, where the number of neurons can be significantly larger, without succumbing to the curse of dimensionality. Addressing these concerns by incorporating a wider range of simulation scenarios and offering a detailed comparative analysis with established methods would significantly strengthen the manuscript, enhancing the credibility and perceived utility of the proposed methods within the neuroscience research community.

For the real data application, the proposed analytical methods to a brief 182-second sleep cycle, analyzing the activity of only ten neurons from each class (pE and pI), again raises concerns about the method's computational demands and scalability. This limited scope suggests that the method may be computationally intensive, potentially restricting its practicality for studies requiring long-term or large-scale neural recordings. Addressing the computational efficiency and scalability of the method is crucial, especially in neuroscience research where understanding complex neural networks often involves analyzing data from dozens to hundreds of neurons over longer periods. Without addressing these concerns, the method's potential impact and adoption within the neuroscience community could be significantly limited.

**Have the authors made all data and (if applicable) computational code underlying the findings in their manuscript fully available?**

Reviewer #2: Yes

Reviewer #3: None

PLOS authors have the option to publish the peer review history of their article (what does this mean?). If published, this will include your full peer review and any attached files.

Reviewer #2: No

Reviewer #3: No

Figure Files:

Data Requirements:

Reproducibility:

References:

---

## [Editor Report · Decision Letter 2]

20 May 2024

Dear Prof. Babadi,

We are pleased to inform you that your manuscript 'Adaptive modeling and inference of higher-order coordination in neuronal assemblies: a dynamic greedy estimation approach' has been provisionally accepted for publication in PLOS Computational Biology.

Best regards,

Daniele Marinazzo

Section Editor

PLOS Computational Biology

Daniele Marinazzo

Section Editor

PLOS Computational Biology

---

## [Editor Report · Acceptance letter]

23 May 2024

PCOMPBIOL-D-23-01661R2 

Adaptive modeling and inference of higher-order coordination in neuronal assemblies: a dynamic greedy estimation approach

Dear Dr Babadi,

I am pleased to inform you that your manuscript has been formally accepted for publication in PLOS Computational Biology. Your manuscript is now with our production department and you will be notified of the publication date in due course.

With kind regards,

Lilla Horvath
